# Starch and Sucrose Metabolism and Plant Hormone Signaling Pathways Play Crucial Roles in *Aquilegia* Salt Stress Adaption

**DOI:** 10.3390/ijms24043948

**Published:** 2023-02-16

**Authors:** Lifei Chen, Yuan Meng, Yun Bai, Haihang Yu, Ying Qian, Dongyang Zhang, Yunwei Zhou

**Affiliations:** College of Horticulture, Jilin Agricultural University, 2888 Xincheng Street, Changchun 130118, China

**Keywords:** *Aquilegia vulgaris*, salt stress, RNA sequencing, starch and sucrose metabolism, plant hormone signal transduction

## Abstract

Salt stress is one of the main abiotic stresses that strongly affects plant growth. Clarifying the molecular regulatory mechanism in ornamental plants under salt stress is of great significance for the ecological development of saline soil areas. *Aquilegia vulgaris* is a perennial with a high ornamental and commercial value. To narrow down the key responsive pathways and regulatory genes, we analyzed the transcriptome of *A. vulgaris* under a 200 mM NaCl treatment. A total of 5600 differentially expressed genes were identified. The Kyoto Encyclopedia of Genes and Genomes (KEGG) analysis pointed out that starch and sucrose metabolism and plant hormone signal transduction were significantly improved. The above pathways played crucial roles when *A. vulgaris* was coping with salt stress, and their protein–protein interactions (PPIs) were predicted. This research provides new insights into the molecular regulatory mechanism, which could be the theoretical basis for screening candidate genes in *Aquilegia*.

## 1. Introduction

It is inevitable for plants to encounter environmental stresses during their life cycles. Of the many conditions of stress, salt stress is one of the main abiotic stresses that affect plant growth negatively [1]. Saline soil makes up 6% of the worldwide soil area, and severe plant damage has been found to be the result of consistent soil-salt accumulation [2]. Elaborating on the molecular regulatory mechanism of salt resistance has great significance for crop loss remission, landscape ecology maintenance and soil-erosion prevention. *Aquilegia vulgaris* (Ranunculaceae) is a widespread and adaptable perennial with a high ornamental and commercial value [3]. Meanwhile, *Aquilegia* is an emerging model plant at the phylogenetic midpoint between *Arabidopsis* and *Oryza* [4], and the tolerance mechanism of *Aquilegia* could provide a theoretical basis for genetic evolution research. In 2018, the whole genome of *A. coerulea* “Goldsmith” was sequenced, which provided powerful data support for transcriptome analyses [5]. In previous research, intensive studies were carried out on pollination biology [6], anthocyanin regulation [7], floral organ morphogenesis [7,8], genetic evolution [9], medicinal components [10] and physiological stress responses [11]. However, the molecular regulatory mechanism in *Aquilegia* under salt stress is poorly understood. Since the ecological quality of the landscape has been damaged due to global climate and soil volatility, revealing the adaptive molecular mechanism for salt stress can further promote the application of *Aquilegia*.

Under salt stress conditions, a high concentration of Na^+^ leads to the disruption of a plant’s osmotic equilibrium and physiological drought, and it is the reason for photosynthesis inhibition, tissue damage and plant death [12]. Starch is not only the main energy source for plants but is also involved in the abiotic stress response mechanism [13,14,15]. With catalysis by β-amylase (BAM), starch can be degraded into small carbohydrate molecules. This process releases energy and maintains stability in the osmotic system in plant cells. In addition, the storage forms of sugar can be catalyzed by β-glucosidases (BGLUs) into different complexes that sustain plant physiology stability [16]. Trehalose is an intermediate product of starch and sucrose metabolism, which plays a vital role in damage reduction [17]. Previous studies have illustrated that carbohydrate metabolism has become a decisive factor for salt stress tolerance [18]. However, the clues are still scattered at present; for instance, is there any other enzyme that is correlated with salt resistance, with the exception of BAM? What is the mechanism by which carbohydrates regulate plant growth and development? These questions have been poorly investigated; in addition, different species have various regulatory mechanisms, and more sufficient evidence and supporting theories are needed [18].

Plant hormones, including indole-3-acetic acid (IAA), abscisic acid (ABA), cytokinin (CK), ethylene (ETH), gibberellin (GA) and brassinolide (BR), play crucial roles in plant growth regulation and stress resistance [1]. Fluctuations in endogenous hormone levels are stimulated by stress conditions; signaling pathways can be activated by a combination of hormones and receptors. Subsequently, physiological and biochemical processes are regulated downstream [19]. The accumulation of ABA leads to the closure of stomata cells and alleviates damage during photosynthesis, which directly adjusts plant growth and compound transportation [20]. However, when induced by stress, the concentration and distribution of IAA can be readjusted. This is highly correlated with root morphogenesis and growth situations [21]. DELLAs, plant growth inhibitors, are recruited by downstream genes in the GA signaling pathway and play essential roles in plant stress adaption [1].

The intermediate products of starch and sucrose metabolism include glucose, sucrose and trehalose 6-phosphate. They can interact with other signaling pathways to adjust downstream responses, such as osmotic regulating substances and signaling molecules [22,23,24,25,26]. Glucose is associated with the expressions of IAA signaling pathway genes, such as *YUCCA*, *TIR1*, *AXR2* and *AXR3*, which regulate plant cell proliferation, expansion and differentiation [27]. Additionally, both synergistic and antagonistic effects have been found between glucose and the CK signaling pathway [28]. The synthesis and transduction of GA are correlated with glucose [29]. Moreover, it has been clarified that *SnRK1* is the key gene in the interaction between glucose and ABA signaling pathways [30]. *Arabidopsis thaliana* with mutation genes in the sugar signaling pathway have incomplete ABA signaling pathways, and individuals with mutation genes that affect ABA synthesis are not sensitive to glucose [31]. These results demonstrate that sugar signaling and plant hormones regulate and control plant growth together. Nevertheless, more research is needed regarding the intermediate processes of the sugar signaling pathway, such as the determining targets of enzymes and the association between microscopic and macroscopic regulatory processes in plants [25].

To explore the responsive mechanism under salt stress, a transcriptome library was constructed after a 200 mM NaCl treatment in *A. vulgaris*. Salt-stress-responsive genes in key regulatory pathways were screened and analyzed based on the RNA-seq database. Our study provides new insights into the molecular regulatory mechanism of *A. vulgaris* under salt stress. Furthermore, the molecular mechanism shows more possibilities for salt-tolerant molecular breeding in *Aquilegia*.

## 2. Results

### 2.1. RNA-Sequencing Quality

A total of 12 transcriptome libraries, at least 41,521,502 clean reads and 6.23 G clean bases, were obtained (Table 1). The Q30 values exceeded 94.81%, and the GC values were between 41.51% and 42.72%. A total of 28,088 de novo genes were assembled (Figure 1), and the contrast ratio between the transcriptome and reference genome exceeded 90.59% (Table 2). The R^2^ among the three samples in the same treatment exceeded 0.85 (Figure 1). According to the principal component analysis (PCA) (Figure 2B), the samples in different treatments were obviously separated, and the samples in the same treatment were aggregated, which indicates that there was a considerable correlation among the repetitions.

### 2.2. Differentially Expressed Genes (DEGs)

A total of 5600 differentially expressed genes (DEGs) were found in this research, including 2550 upregulated genes and 3050 downregulated genes. In total, 1979, 918 and 2703 genes were differently expressed after salt treatment for 12 h, 24 h and 48 h, respectively (Figure 3A). Moreover, there were more downregulated genes than upregulated genes in each time node (Figure 3F–H). The numbers of differently expressed genes were acquired in the comparisons of 12 h and 24 h, 12 h and 48 h, and 24 h and 48 h (Figure 3B–D, respectively). A total of 188 DEGs were shared in the three comparisons (Figure 3E). These results show that the genes responded differently in each time node, and 188 genes were found to play roles throughout the salt treatments.

### 2.3. Gene Ontology (GO) Enrichment Analysis

Based on the screening condition of *p* value ≤ 0.05, a total of 8355 genes were annotated as significantly enriched. Fatty acid biosynthetic process, microtubule cytoskeleton and O-acyltransferase activity were the main items that were enriched after 12 h (Figure 4A and Appendix A). Metal ion transport, extracellular region and transmembrane transporter activities were the main items that were enriched after 24 h (Figure 4B and Appendix A). Peptide metabolic process, ribosome and structural constituent of ribosome were the main items that were enriched after 48 h (Figure 4C and Appendix A).

For further investigation, the genes whose Gene Ontology (GO) annotations were involved in starch and sucrose metabolism and plant hormone signal transduction were collected. After 12 h, 187 genes were enriched, such as polysaccharide metabolic process, cell wall and external encapsulating structure compound. Moreover, 109 genes were enriched after 24 h, such as signal transduction, signaling, cell communication and carbohydrate catabolic process. However, only six genes were enriched after 48 h, including amylase activity. In this case, the expressions of the genes involved in starch and sucrose metabolism and plant hormone signal transduction were more active at 12–24 h than at 48 h.

### 2.4. Kyoto Encyclopedia of Genes and Genomes (KEGG) Enrichment Analysis

A total of 13 Kyoto Encyclopedia of Genes and Genomes (KEGG) pathways were significantly enriched based on the screening condition of Padj ≤ 0.05 for the investigation of the gene expressions in the key pathways, in which 2197 differentially expressed genes were found. Four, eight and one pathways and 678, 363 and 1156 genes were enriched significantly after salt treatment for 12 h, 24 h and 48 h, respectively (Figure 5). The mainly enriched KEGG pathways were starch and sucrose metabolism, plant hormone signal transduction and ribosome after 12 h, 24 h and 48 h, respectively.

### 2.5. Gene Expression Analysis in Starch and Sucrose Metabolism Pathway

A total of 74 genes were found in the starch and sucrose metabolism pathway (Figure 6). Six genes were up- or downregulated in the trehalose synthesis process after salt treatment for 12 h (Figure 6A), which shows that trehalose was involved in the salt tolerance mechanisms of *A. vulgaris*. The starch synthase genes (*SS3*, *SS4* and *GBSS1*) in the starch synthesis process (Figure 6B) were downregulated after salt treatment for 12 h, upregulated at 24 h and then tended to stabilize. The 1,4-alpha-glucan branching enzyme gene (*EMB2729*) reached the peak at 12 h and then decreased, which indicates that starch synthesis was enhanced instead of amylose. In the process of starch hydrolysis (Figure 6D), *ISA1*, *ISA3*, *CT-BMY*, *DPE2* and *PHS2* reached the peak at 12 h and then decreased at 24 h; *BAM7*, *DBE1*, *AT5G11720* and *PHS1* decreased at 12 h and then reached the peak at 24 h. Thus, the salt treatment induced amylolysis during the 12–24 h period and then abated.

In the process of sucrose synthesis and hydrolysis (Figure 6C), *SPS1F*, *ATPS4F* and *SPS3F* reached the peak at 24 h and then decreased, which indicates that the 24 h salt treatment induced sucrose synthesis in *A. vulgaris*. Regarding the sucrose synthase genes (*SUS3* and *SUS6*) and β-fructofuranosidase genes (*ATBETAERUCT4*, *AT1G62660* and *AT1G12240*), which were annotated in the process of sucrose hydrolysis, their expressions increased and then decreased, with peaks at 12 h. The gene that was annotated as α-glucosidase (*AT5G1172*) reached the lowest value at 12 h and then increased substantially. That is to say, the dominant genes that regulated sucrose hydrolysis varied with the salt treatment process.

A total of 26 genes were annotated as the beta-glucosidase (*BGLU*) family in cellulose hydrolysis (Figure 6E), of which 12 genes (*BGLU1*, *BGLU46*, *BGLU10*, *BGLU11* and *BGLU1*) reached the peak at 12 h. A total of 10 *BGLU* genes (*BGLU40*, *BGLU10*, *BGLU44*, *BGLU11* and *BGLU45*) and 1 endoglucanase gene (*GH9A1*) increased substantially at 24 h, and 8 genes (*BGLU10*, *BGLU11* and *BGLU45*) reached the peak. A total of four genes (*BGLU10*, *BGLU40*, *BGLU17* and *AT5G20950*) were upregulated stably with a peak at 48 h. In general, the expressions of the genes that regulated the starch and sucrose metabolism pathway varied with the salt treatment process, and these results reveal that the energy source distribution was affected by the salt treatment in *A. vulgaris*.

### 2.6. Gene Expression in Plant Hormone Signal Transduction

A total of 87 genes were annotated in the plant hormone signal transduction pathways (Figure 7). In the IAA, CK and GA signaling pathways, 50, 7 and 6 genes were found, respectively. The expressions of 37 genes decreased at 12 h and then increased at 24 h in the IAA signaling pathway (Figure 7A), of which 19 genes (*TIR1*, *IAA16*, *SHY2*, *ETT*, *At2G21210*, *AT4G34760*, *AT4G34770*, *AT1G56150*, *AT3G43120*, *AT5G18060* and *AT4G38840*) and 15 genes (*LAX2*, *LAX1*, *TIR1*, *IAA19*, *AT4G34770*, *AT2G21210*, *AT2G21220* and *AT4G38840*) reached the lowest values at 12 h and 48 h, respectively. The expressions of four genes (*ARR6*, *ARR9* and *RR3*) in the CK signaling pathway (Figure 7B) decreased at 12–24 h, with a small increase at 48 h. Induced by the salt treatment, the genes’ expressions in the GA signal transduction pathway (Figure 7C) changed in a complicated manner; the expressions of two *DELLA*s (*GAI*) maintained a level lower than that at 0 h after 12 h of downregulation. Thus, IAA, CK and GA signaling were inhibited by the salt treatment.

Under salt stress, the ABA, BR and JA signaling pathways were involved in 11, 7 and 6 genes, respectively (Figure 7). Firstly, the expression of five genes increased and then decreased in the ABA signal transduction pathway (Figure 7D), of which four *SnRK2* family genes reached the peak at 12 h. Two *ABF*s, which belonged to the downstream genes of *PP2C*, reached the lowest values at 12 h and then recovered to varying degrees. In addition, the expression of the *CYCD3* gene increased and then decreased, except for upstream of BSK in the BR signaling pathway (Figure 7E). The responsive genes in the JA signaling pathway changed differently under the salt treatment (Figure 7F). These results show that the ABA signaling pathway was negatively affected by the initial salt treatment but was enhanced in anaphase, which is opposite to the case of the BR signaling pathway, and the complex defense system of *A. vulgaris* was constituted together with the JA signaling pathway.

### 2.7. Interaction Network Analysis

A total of 155 protein–protein interaction (PPI) IDs were uploaded to the Sting online database, and they were involved in the starch and sucrose metabolism pathway and the plant hormone signal transduction pathway. The result shows that 67 proteins in the starch and sucrose metabolism pathway formed an interaction network with 49 proteins in the plant hormone signal transduction pathway (Figure 8). Furthermore, we found that CT-BMY interacted with IAA19 and that BGLU40 interacted with the DELLA family.

### 2.8. Data Reliability Analysis with Quantitative Real-Time PCR

In order to verify the reliability of the RNA-seq, the relative expressions and log 2-fold changes of 16 genes were calculated based on Ct. The upregulation or downregulation trends in the quantitative real-time PCR (qRT-PCR) were the same as those in the RNA-seq (Figure 9), which indicates the considerable reliability of the RNA-seq.

## 3. Discussion

Soil salination has become a global issue that strongly restricts crop production and landscape ecological quality. *A. vulgaris* has a great potential ornamental and commercial value in northern hemisphere countries [3]. The transcriptome of *A. vulgaris* under salt treatment was analyzed in this research. A total of 5600 DEGs were found, and starch and sucrose metabolism and plant hormone signal transduction were significantly enriched in the KEGG analysis, of which the genes were the focus of subsequent investigations.

To cope with abiotic stress, secondary metabolites in the glycogroup form are synthesized in plants. After glycosylation, these secondary metabolites are more soluble and stable than they were before. The synthesis process is catalyzed by the β-glucosidase (BGLU) family, which plays an essential role in cell membrane stability and osmotic potential balancing [16]. In addition, the functions of BGLUs also include cell wall degradation, lignification and signal transduction [32]. A total of 29 *BGLU*s were obtained in the KEGG enrichment analysis, and they were the most numerous in the starch and sucrose metabolism pathway. Previous studies have illustrated a diverse mode of action of the *BGLU* family in different species; for example, the upregulation of *BGLU*s in *Solanum lycopersicum* was induced by salt stress [33], and the *BGLU*s in *Medicago truncatula* underwent various changes under salt stress [16]. The functions of some *BGLU*s have been verified; for instance, in a previous study, *CsBGLU12* was induced by salt treatment, during which the antioxidant flavonol content of over-expressed *Nicotiana benthamiana* accumulated and alleviated the stress damage level [34]. However, mutant *A. thaliana* plants, in which the salt-response genes *AtBGLU1* and *AtBGLU19* were knocked out, have been found to be less sensitive to salt stress [35]. In addition, BGLU belongs to the GH1 family, which has been an ideal material for plant engineering [35]. In this study, the BGLU family was induced to catalyze glycosylated osmotic regulation in order to protect the cell membrane system’s integrity, which was the salt stress adaptive strategy in *A. vulgaris*.

Plant growth, development and energy distribution are regulated by plant hormone signaling in stress adaption [36]. A total of 28 genes were found in the plant hormone signaling pathways. Among them, five *SAUR*s were inhibited by the salt treatment at all time points, and this result is different from that of previous studies [37,38,39]. It has been elucidated that IAA regulates plant morphogenesis under stress conditions, including vegetative and reproductive growth [2,40,41,42]. IAA, Transport Inhibitor Response 1 (TIR1) and the auxin signal transduction-related protein (AFB) combine to form a complex, which leads to the ubiquitination and degradation of AUX/IAA, restrains the function of auxin response factors (ARFs) and negatively affects the expressions of the downstream transcription factors small auxin-upregulated RNAs (SAURs) [43]. Plant cell division and expansion are regulated by SAURs and correlated with plant morphogenesis [44]. The over-expression of *SAUR* in *A. thaliana* leads to cell elongation [45], and *SAUR* has also been found to be associated with the stem growth of sunflowers (*Helianthus annuus*) [46]. Depending on the IAA signaling pathway, a restrained expansion of the leaf area and reduced nutrient consumption were formed as a survival strategy under salt stress in *A. vulgaris*.

According to the protein–protein interaction (PPI) analysis, a protein interaction network was found among the starch and sucrose metabolism and plant hormone signal transduction pathways. The result shows that CT-BMY and the AUX/IAA protein IAA19 interacted with each other. AUX/IAA plays a crucial role in the IAA signaling pathway, and it is correlated with plant embryonic development, lateral root growth and floral development [47]. In 2001, Purgatto first discovered that IAA affected the activation of *BAM* during banana fruit ripening, which delayed starch degradation [48]. Glucose is one of the starch degradation products, and it provides a substrate for the biosynthesis of resistant active substances [49]. Thus, starch hydrolysis, which is regulated by the IAA signaling pathway, was one of the survival tactics in *A. vulgaris*; further studies are needed to determine their interaction relationship. Moreover, according to the PPI network, BGLU40 interacted with DELLA, which belongs to the GA signaling pathway. It has been illustrated that a glycoside bond can be hydrolyzed by BGLU, release IAA and ABA combined with glucose and endow the hormone with biological activities [32]. The promoters of the *PtBGLU* family contain the ABA-responsive element ABRE, and the gene members can respond to stress conditions mediated by the ABA signaling pathway [50]. Moreover, the *OsBGLU* family is also correlated with the IAA and ABA signaling pathways [32]. GA plays an important role in plant growth and development. DELLA belongs to the GA signaling pathway; it is a growth inhibitor that can regulate downstream transcription factors, and it is related to the key proteins that affect plant growth [51]. DELLA degradation is the central adjustment switcher in the GA signaling pathway [52]. In 2013, Paparelli [29] demonstrated that there were diurnal changes in starch synthesis and degradation; starch was accumulated by photosynthesis during the daytime, and GA synthesis and plant growth were powered by starch degradation during the nighttime. In addition, the regulation factor O-Linked N-acetylglucosaminyltransferase (OGT), which is independent of the GA signaling pathway in *A. thaliana*, could catalyze the synthesis of O-linked N-acetylglucosamine (O-GlcNac). DELLA glycosylate was catalyzed by O-GlcNac; therefore, non-GA mediated post-modification was determined to be a considerable way to affect DELLA activity [53]. In 2019, it was found that DELLA and BGLU were both significantly enriched during floral organ development in *Dimocarpus longan* [54]; however, there was no experimental evidence to prove the interaction between them. Although the circadian rhythm in this research was inevitable, the expression of DELLA was inhibited throughout the entire treatment time, which suggests that it was predominantly negatively affected by the salt treatment. We further speculated that BGLU40 is correlated with the post-modification of DELLA and with the regulation of DELLA and downstream transcription factors’ activities; thus, it could represent a salt stress adaption strategy in *A. vulgaris*.

## 4. Materials and Methods

### 4.1. Plant Materials and Salt Treatment

The seeds of *A. vulgaris* were harvested from the ornamental resource greenhouse of Jilin Agricultural University (125°43′ E, 43°82′ N), Changchun City, Jilin Province, China. Then, all seeds were sown into 6.5 × 6.5 × 9 cm^3^ plastic pots filled with turf. When the seedlings had six leaves, each pot was irrigated with 50 mL of a 200 mM NaCl solution, according to the pre-experiment. A control test method was adopted in this study. After treatment for 0 h (control), 12 h, 24 h and 48 h, the six fully expanded leaves were taken as the material for RNA extraction, and they were collected and rapidly frozen in liquid nitrogen and stored under −80 °C conditions. At least 500 mg of tissue was collected in each treatment, and each treatment was repeated three times.

### 4.2. RNA Extraction and cDNA Library Construction

A standard process was taken as the method for RNA extraction [55]. The mRNAs with poly-A tails were enriched using Oligo (dT) magnetic beads and then broken randomly in a fragmentation buffer. The mRNA fragments were taken as the templates for cDNA transcription after the terminals were repaired and the A tails were added, and the libraries were obtained using PCR amplification with the specific primers. Moreover, the libraries were sequenced using the Illumina Novaseq platform (Illumina, San Diego, CA, USA).

### 4.3. Reference Genome Alignment

The genome sequence of *A. coerulea* “Goldsmith” was obtained from NCBI (https://ftp.ncbi.nlm.nih.gov/genomes/all/GCA/002/738/505/GCA_002738505.1_Aquilegia_coerulea_v1/, accessed on 10 October 2020). Then, index preparation and a genome comparison were conducted using HISAT2 [56] accurately.

### 4.4. Sample Correlation and Gene Expression Analysis

The correlation between the samples was determined using Pearson correlation (R^2^), and the principal component analysis (PCA) was calculated using the method of linear algebra. The read number, which was mapped to each gene, was analyzed using featureCounts (1.5.0) [57]. FPKM (the expected number of fragments per kilobase of transcript sequence per millions of base pairs sequenced) was calculated based on the length of the gene and the read count mapping to the gene. The expression difference was analyzed using DESeq (1.20.0) [58], and Padj was adjusted using the Benjamini and Hochberg method [59] to control the error discovery rate. After correction, Padj ≤ 0.05 and |Log 2 fold change| ≥ 1 were considered as the thresholds for significantly differentially expressed genes. A heat map was visualized using TBtools (1.108) (South China Agricultural University, Guangzhou, China) after the FPKM values were normalized by taking the log base 2.

### 4.5. Enrichment and Interaction Analyses

Gene Ontology (GO) and Kyoto Encyclopedia of Genes and Genomes (KEGG) enrichment analyses were conducted using the clusterProfiler R package (3.4.4) [60]. The corrected *p* value ≤ 0.5 was selected as a threshold condition for the significant enrichment of differential genes in the GO terms. The KEGG enrichment was conducted as the reference data of *A. thaliana*. Protein–protein interactions (PPIs) were analyzed using String (https://cn.string-db.org/) accessed on 8 February 2021, and the network was visualized using Cytoscape (3.9.0) (National Institute of General Medical Sciences, Bethesda, MD, USA).

### 4.6. Quantitative Real-Time PCR (qRT-PCR)

RNA extraction was conducted as described above, and RNA reverse transcription was conducted using a cDNA Synthesis Kit (All-in-one 1st Strand cDNA Synthesis SuperMix, Novoprotein, Suzhou, China). Sixteen genes were randomly selected, and their Ct values were measured using the qTOWER3 detecting system (Analytik, Jena, Germany) with SYBR qPCR SuperMix Plus (E096, Novoprotein, Suzhou, China). As shown in Appendix A, the primers were designed using Primer Premier 5 (Primier Biosoft International, Palo Alto, CA, USA). Moreover, the internal control gene was *IPP2* as reported in [8]. The relative expressions were calculated using the 2^−ΔΔCt^ method [61]. The Log 2-fold change was calculated by comparing the salt treatment group with the control group (0 h) using the method reported in [62].

## 5. Conclusions

The *A. vulgaris* transcriptome was sequenced under a 200 mM NaCl treatment for 0 h, 12 h, 24 h and 48 h. A KEGG enrichment analysis revealed that the starch and sucrose metabolism pathway and the plant hormone signal transduction pathway played vital roles during the salt treatment. In addition, glycoconjugate biosynthesis regulated by the BGLU family could balance the osmotic potential and maintain cell membrane stability. The IAA signaling pathway to slow down growth and reduce energy consumption was formed as one of the survival strategies in *A. vulgaris*. Moreover, the starch and sucrose metabolism pathway and the plant hormone signal transduction pathway were combined to constitute the defense system in *A. vulgaris*. The application of the *BGLU* family in plant engineering and the starch degradation mechanism mediated by the IAA signaling pathway need to be further studied. This research provides new insights into the stress resistance in *Aquilegia* and puts forward more possibilities for the excavation of salt-tolerant genes.

## Figures and Tables

**Figure 1 ijms-24-03948-f001:**
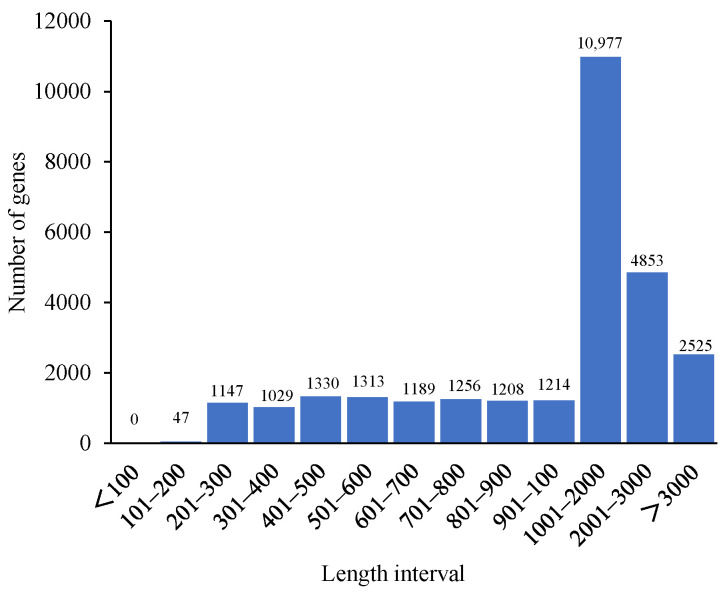
The lengths of the genes.

**Figure 2 ijms-24-03948-f002:**
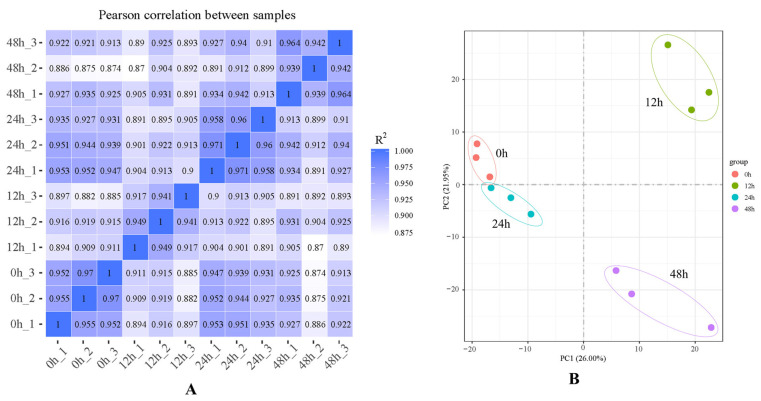
Correlation analysis and principal component analysis (PCA): (**A**) sample correlation analysis; (**B**) sample distributions in PCA 1 and PCA 2.

**Figure 3 ijms-24-03948-f003:**
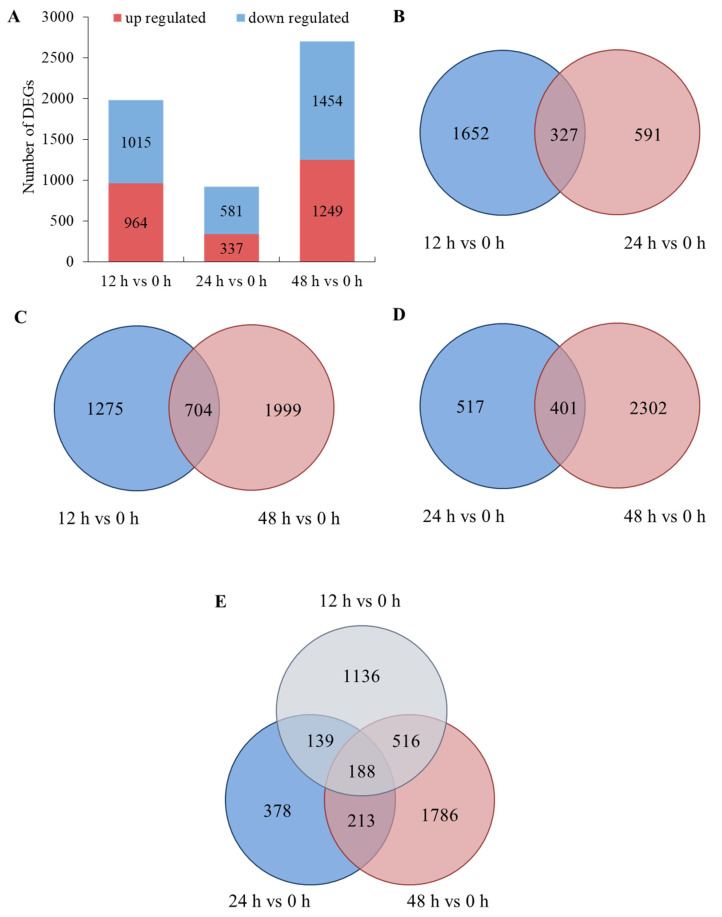
Differentially expressed genes (DEGs): (**A**) numbers of DEGs in each time node; (**B**) Venn diagrams for 12 h vs. 0 h and 24 h vs. 0 h; (**C**) Venn diagrams for 12 h vs. 0 h and 48 h vs. 0 h; (**D**) Venn diagrams for 24 h vs. 0 h and 48 h vs. 0 h; (**E**) Venn diagrams for 12 h vs. 0 h and 24 h vs. 0 h and 48 h vs. 0 h; (**F**) volcano plots for 12 h vs. 0 h; (**G**) volcano plots for 24 h vs. 0 h; (**H**) volcano plots for 48 h vs. 0 h.

**Figure 4 ijms-24-03948-f004:**
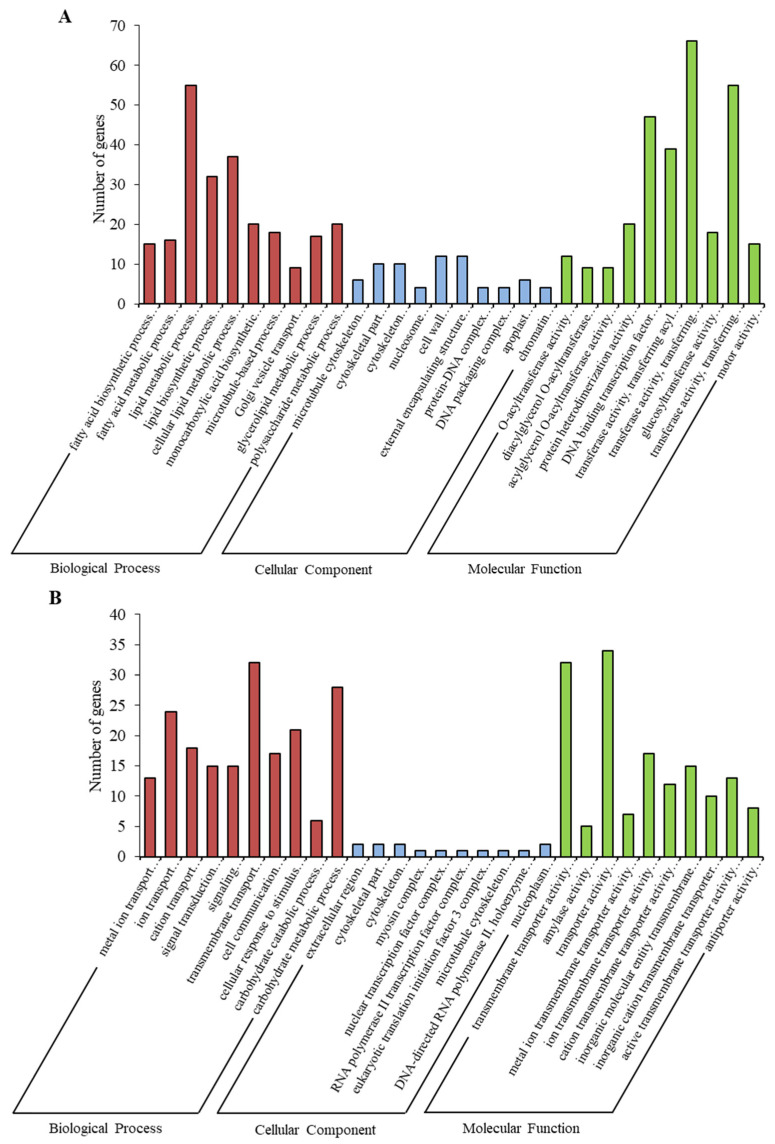
Gene Ontology (GO) enrichment analysis in the following groups: (**A**) 12 h vs. 0 h; (**B**) 24 h vs. 0 h; (**C**) 48 h vs. 0 h.

**Figure 5 ijms-24-03948-f005:**
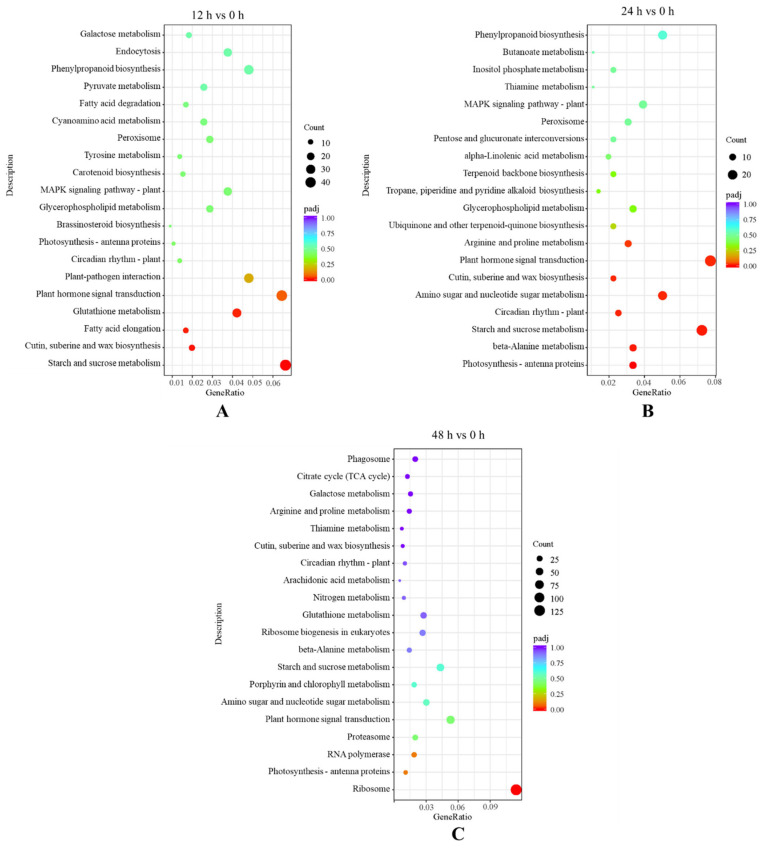
Kyoto Encyclopedia of Genes and Genomes (KEGG) enrichment analysis in the following groups: (**A**) 12 h vs. 0 h; (**B**) 24 h vs. 0 h; (**C**) 48 h vs. 0 h.

**Figure 6 ijms-24-03948-f006:**
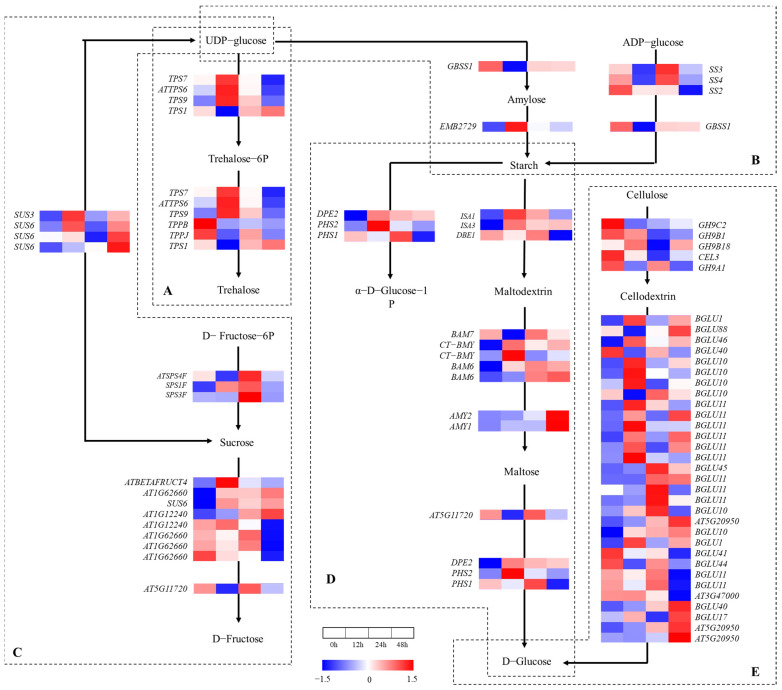
Heat map of annotated genes in starch and sucrose metabolism pathway. Solid lines and arrows represent biological processes and directions, while areas separated by dotted lines represent different types of biological processes. (**A**) Trehalose synthesis process; (**B**) starch synthesis process; (**C**) sucrose synthesis and hydrolysis process; (**D**) starch hydrolysis process; (**E**) cellulose hydrolysis process.

**Figure 7 ijms-24-03948-f007:**
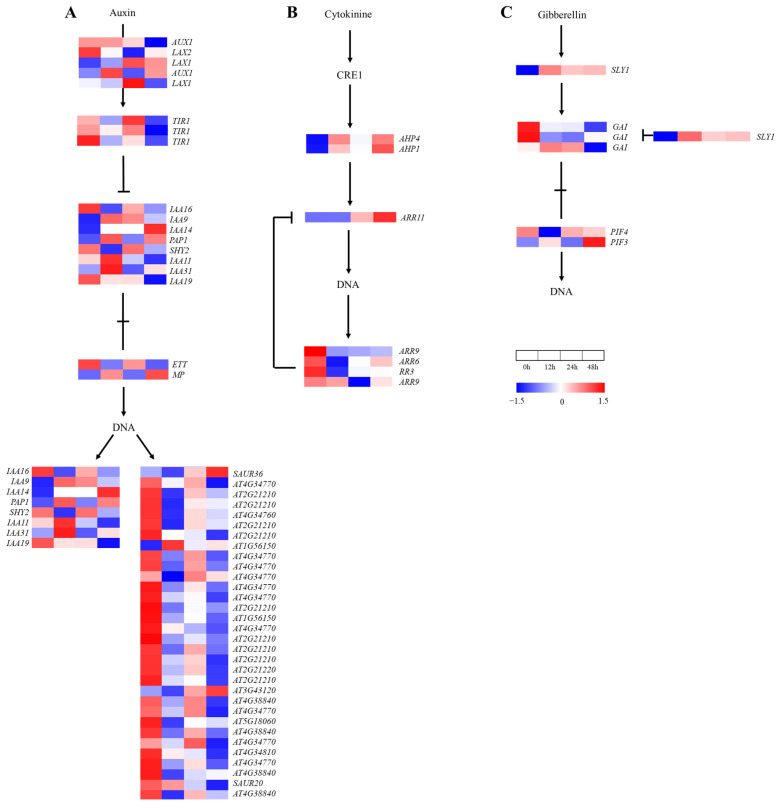
Heat map of annotated genes in plant hormone signal transduction pathway. Solid lines and arrows indicate the signal transduction process and direction. (**A**) Indole-3-acetic acid (IAA) signaling pathway; (**B**) cytokinin (CK) signaling pathway; (**C**) gibberellin (GA) signaling pathway; (**D**) abscisic acid (ABA) signaling pathway; (**E**) brassinolide (BR) signaling pathway; (**F**) jasmonic acid (JA) signaling pathway.

**Figure 8 ijms-24-03948-f008:**
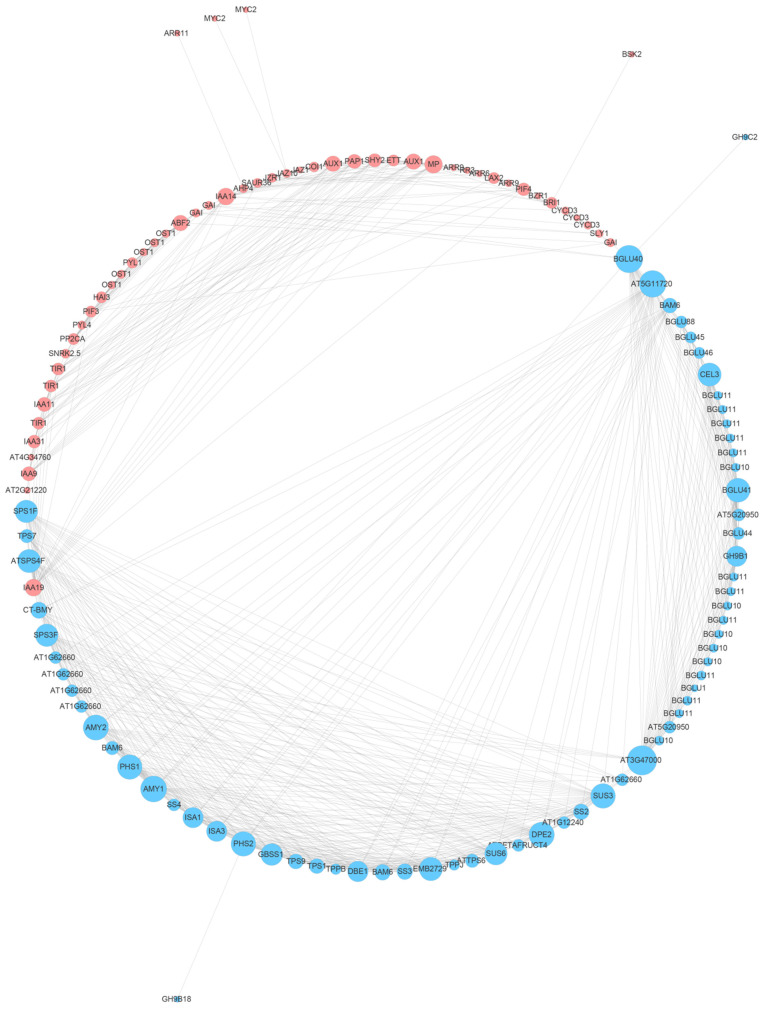
Protein–protein interaction (PPI) network among the starch and sucrose metabolism pathway and the plant hormone signal transduction pathway. The blue and red nodes represent the sucrose metabolism pathway and the plant hormone signal transduction pathway, respectively. The diameters of the nodes represent the interaction frequency.

**Figure 9 ijms-24-03948-f009:**
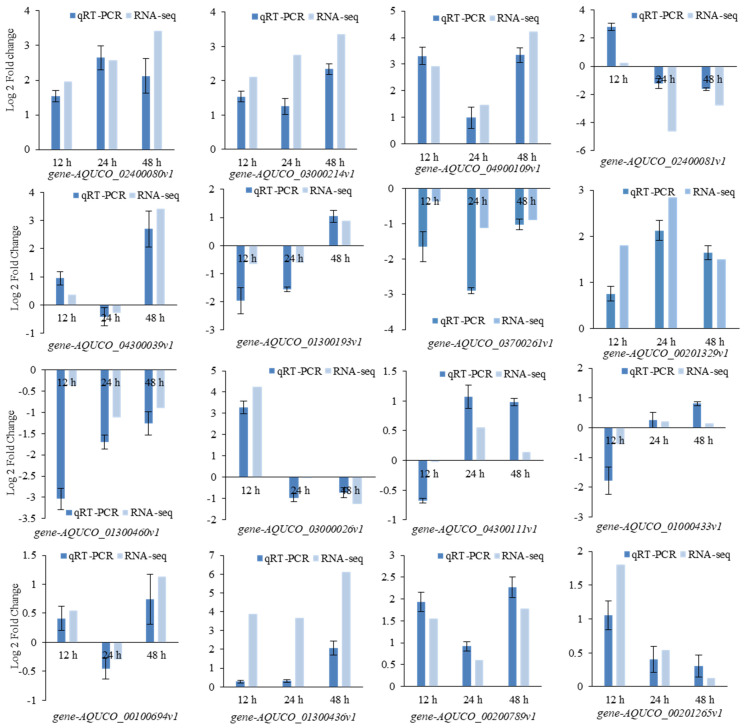
Log 2-fold changes of 16 genes in quantitative real-time PCR (qRT-PCR) and RNA-seq.

**Table 1 ijms-24-03948-t001:** RNA-sequencing quality.

Sample	Raw Reads	Clean Reads	Clean Bases	Error Rate	Q20	Q30	GC PCT
0 h_1	47,647,814	46,506,466	6.98 G	0.02	98.37	94.88	42.31
0 h_2	47,064,990	45,853,912	6.88 G	0.02	98.41	94.96	42.58
0 h_3	46,642,716	45,494,096	6.82 G	0.02	98.48	95.14	42.72
12 h_1	42,376,174	41,521,502	6.23 G	0.02	98.34	94.87	42.23
12 h_2	47,123,344	45,962,436	6.89 G	0.02	98.34	94.91	42.05
12 h_3	49,174,170	47,628,388	7.14 G	0.02	98.48	95.19	41.85
24 h_1	45,614,350	44,336,182	6.65 G	0.02	98.54	95.31	42.25
24 h_2	46,115,956	44,862,266	6.73 G	0.02	98.5	95.13	42.31
24 h_3	51,810,052	50,770,432	7.62 G	0.02	98.55	95.26	42.29
48 h_1	45,223,922	44,376,174	6.66 G	0.02	98.43	95.14	42.09
48 h_2	47,939,292	46,686,300	7.0 G	0.02	98.3	94.81	41.51
48 h_3	48,060,240	46,931,440	7.04 G	0.02	98.41	95.1	42.2

**Table 2 ijms-24-03948-t002:** A comparison of each sample with the reference genome.

Sample	Total Reads	Total Map	Multi Map
0 h_1	46,506,466	91.08%	2.20%
0 h_2	45,853,912	91.21%	2.24%
0 h_3	45,494,096	92.07%	2.26%
12 h_1	41,521,502	91.56%	2.42%
12 h_2	45,962,436	91.05%	2.36%
12 h_3	47,628,388	90.82%	2.54%
24 h_1	44,336,182	91.62%	2.37%
24 h_2	44,862,266	91.86%	2.32%
24 h_3	50,770,432	91.77%	2.42%
48 h_1	44,376,174	91.15%	2.50%
48 h_2	46,686,300	90.59%	2.52%
48 h_3	46,931,440	91.46%	2.53%

## Data Availability

The data presented in this study are openly available in the Sequence Read Archive (SRA) database in NCBI under accession number PRJNA931641.

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
