# Peer review of "Starch and Sucrose Metabolism and Plant Hormone Signaling Pathways Play Crucial Roles in Aquilegia Salt Stress Adaption"

_ijms, 2023, doi:10.3390/ijms24043948_

Round 1
Reviewer 1 Report
I suggest the authors need to more work to improve the sections of the manuscript indicated on the file uploaded herewith my comments:
Title:
RNA-seq reveals starch sucrose metabolism and plant hormone signaling pathways play crucial roles in Aquilegia salt stress adaption
General comments
Authors should try all their best to significantly improve the English language of the Manuscript.
Misspellings, many types of mistakes and all other sorts of errors in the paper should be carefully rechecked and recorrected.
Abstract
Line 13: please define the abbevations GO and KEGG
1. Introduction
Authors need to improve the English language of this section.
Line 23: please correct the words in the sentence “Salinity soil was 6% proportion of the worldwide soil area, sever plant damage was “to is……severe”
Objectives of the study is not provided
3. Results
Figure 2. not clear readable
Figure 3. not clear readable
Figure 4. very difficult to read and it is not clear
Figure 5. very difficult to read and it is not clear
Figure 7. very difficult to read and it is not clear
Figure 8. very difficult to read and it is not clear
Discussion
Authors need to improve the English language of this section.
Materials and Methods
Study area etc was not known
Study type, sample size etc was not suffieciently provided
Data analysis was forgotten or not provided
References
I suggest to remove old references.

Author Response
Dear reviewer,
We are really appreciate for your excellent and professional revision of this manuscript. We have checked the manuscript according to the comments. After carefully studying, we have made corresponding changes on the manuscript and uploaded to the attachment. Hope these will make it more acceptable for publication.
If any other information or modification are needed, please let me know, thank you so much.
Yours sincerely,
Yuan Meng

Reviewer 2 Report
General Comments:
Chen and Meng et al reported their data analysis for “salt stress, one of the main abiotic stresses” for resistance of plant growth. A deeper understanding of the “ornamental plant molecular regulation mechanism from “salt stress” is very important for ecology development of soil salinity area. During their work, they analyzed A. vulgaris salt stress transcriptome under 200mM NaCl treatment. From the screening of 5600 differentially expressed genes, they determined starch and sucrose metabolism, plant hormone signal transduction and lipid metabolic were closely related when A. vulgaris coped with salt stress, involving 74 and 87 genes, respectively. I believe their research is meaningful for discovery and understanding Aquilegia resistance mechanism and theoretical basis, which will assist further genes screening work. However, some revisions are necessary to make this paper more concise, clear and readable. For example, the title and some statement should be clearer and more consistent. Some possible error bars, relationships between different effects are needed. After the authors make these revisions, I believe their manuscript can be considered for publication in Pharmaceutics.
1. The title “RNA-seq reveals starch sucrose metabolism and plant hormone signaling pathways play crucial roles in Aquilegia salt stress adaption” is too long and a little confusing. It is suggested to rewrite it shortly.
2. From the abstract, “lipid metabolic were enriched significantly” However, why the following statement only mentioned “Starch and sucrose metabolism, plant hormone signal transduction” without “lipid metabolic”?
3. “Salinity soil was 6% proportion of the worldwide soil area.” should be “Salinity soil was made up of 6% of the worldwide soil area” or “6% proportion of the worldwide soil area was salinity soil”.
4. “More focuses are needed on the intermediate process of sugar signal pathway, the substrate of enzyme in each reactions and the changes of sugar signal from microscopic to macroscopic regulatory processes in plants” What is the possible relationship between “Starch and sucrose metabolism” and “plant hormone signal transduction”. Are they positively related or negatively related?
5. From Figure 1, 2, and 4, was the error-bar needed for the data analysis? If so, related error-bar should be added.
6. In the conclusion part, “This study provided comprehensive data support for stress resistance in Aquilegia” To what extent, the authors have identified the important roles of “Starch and sucrose metabolism, plant hormone signal transduction”?
Author Response

(The authors gave the same response as above.)
